

# Technical Note: A minimally-invasive experimental system for pCO₂ manipulation in plankton cultures using passive gas exchange (Atmospheric Carbon Control Simulator)

Brooke A. Love[1], M. Brady Olson[1], Tristen Wuori [1]

[1]Shannon Point Marine Center, Western Washington University, 1900 Shannon Point Drive, Anacortes, WA, 98221, U.S.A.

*Correspondence to*: Brooke A. Love (Brooke.Love@wwu.edu)

**Abstract.** As research into the biotic effects of ocean acidification has increased, the methods for simulating these
environmental changes in the laboratory have multiplied. Here we describe the atmospheric carbon control simulator
(ACCS) for the maintenance of plankton under controlled pCO₂ conditions, designed for species sensitive to the physical
disturbance introduced by bubbling of cultures and for studies involving trophic interaction. The system consists of gas
mixing and equilibration components, coupled with large volume atmospheric simulation chambers. These chambers allow
gas exchange to counteract the changes in carbonate chemistry induced by the metabolic activity of the organisms. The
system is relatively low cost, very flexible, and when used in conjunction with semi-continuous culture methods, increases
the density of organisms kept under realistic conditions, increases the allowable time interval between dilutions, and/or
decreases the metabolically driven change in carbonate chemistry during these intervals. It accommodates a large number of
culture vessels, which facilitate multi-trophic level studies and allow the tracking of variable OA responses within and across
plankton populations. It also includes components that increase the reliability of gas mixing systems using mass flow
controllers.



## 1 Introduction

Ocean Acidification (OA) is the process by which increasing atmospheric $CO_2$ leads to increasing dissolved $CO_2$ in marine

waters and the suite of chemical and biological responses driven by this change. Manipulative OA research has increased

dramatically, and while the scientific community has begun to set standards for data acquisition and experimental design

(Cornwall and Hurd, 2015; Dickson et al., 2007; Riebesell et al., 2010), experimental approaches remain diverse because of

the wide scope and variety of questions. While many innovative systems are in use, most are not well suited to relatively

long-term maintenance of organisms sensitive to continuous bubbling, or cannot accommodate a large number of culture

vessels. These are both key characteristics for cross-trophic level or cross generational studies in plankton, e.g. tracking

ontogenesis of crustacean zooplankton populations. In addition, static control of chemistry has been criticized (Wahl et al.,

2015b) and systems which are capable of simulating natural variability in conditions are of particular interest.

The Atmospheric Carbon Control Simulator (ACCS) uses passive gas exchange to simulate the effect of atmospheric

exchange in modulating the effects of photosynthesis and respiration on surface water chemistry without undue physical

disturbance of delicate planktonic organisms. This approach mimics real conditions in surface and near shore waters where

the disequilibrium caused by photosynthesis and respiration is subject to partial re-equilibration through air sea gas

exchange. The ACCS is particularly suited for phytoplankton and zooplankton work because it helps maintain carbonate

chemistry with minimal physical stress, and accommodates numerous culture vessels, which are useful in tracking variability

associated with zooplankton feeding and reproductive metrics. The system is not limited to passive gas exchange for

carbonate chemistry control. Direct bubbling can be used to maintain cultures or to pre-equilibrate media for semi-

continuous cultures (described below). The ACCS is designed to accommodate differing trophic levels and differing levels

of desired carbonate control, re-equilibration, or metabolically driven cycling.

On the broadest scale, experimental approaches to studying OA are classified as microcosms or mesocosms. Mesocosm

studies have increased in number in recent years and are well suited to address questions about OA effects on biological

communities, while work in microcosms has been criticized because small volumes may have increased enclosure effects

and limited realism (Hendriks et al., 2010; Stewart et al., 2013; Wernberg et al., 2012). Although the ACCS is a microcosm

system, and has limited potential for realistic simulation of the complexities of community dynamics in a changing ocean,

microcosms do have a strong role to play in elucidating the underlying mechanisms that drive those dynamics. The detailed

species and interspecies level effects observed in microcosms can be coupled with whole community experiments or

observations of natural systems and enhance the interpretation of both types of study (e.g. Webb et al., 2016).

Many studies of planktonic organisms (particularly autotrophs) employ dilute batch culture, where seawater media pre-

conditioned to the appropriate chemistry is inoculated at a low density, sealed, and allowed to grow for a limited period of





time (Eberlein et al., 2016; Hoins et al., 2016; e.g. Wilcox-Freeburg et al., 2013; Zhang et al., 2015). Changes in water chemistry can exceed 1 pH unit and 15% of DIC during the course of the batch culture when biomass is high or duration is long (Brutemark et al., 2015). Dilute batch culture systems are not well suited for zooplankton studies because cultures are subject to gas exchange each time they are opened for feeding, and dilute cultures of prey can cause encounter rates to limit growth and grazing rates of zooplankton.


A common modification of this batch technique is to bubble $CO_2$/air mixtures directly into the culture vessels (Kremp et al., 2012; Neale et al., 2014; Talmage and Gobler, 2009). This allows $CO_2$ and $O_2$ to re-equilibrate with target concentrations constantly over the course of the experiment. Plankton cultures aerated in this way can achieve somewhat higher densities while still maintaining carbonate chemistry, but this is not appropriate for organisms which are sensitive to turbulence and

the shear stress that is induced by bubbling (Gifford, 1993; Riebesell et al., 2010; Shi et al., 2009; Wolfe et al., 2002). We observed detrimental effects of bubbling, including reduced growth rates and formation of aggregates during culturing of autotrophic dinoflagellates (unpubl. data). Direct bubbling of cultures also dampens or eliminates diurnal cycling in carbonate chemistry, which may not be desirable.

Several systems are in use which modify the carbonate system by active control of the introduction of $CO_2$ into the water, activating $CO_2$ addition only when pH is above the set point for a given treatment (Hoffmann et al., 2013; Isari et al., 2016; MacLeod et al., 2015; e.g. Wilcox-Freeburg et al., 2013). These systems have several advantages, including little to no mechanical stress, and the potential to control diurnal chemical cycles. They may be batch or flow through in nature. One drawback is that each culture vessel usually requires its own independent pH control circuit or valve port, which limits the

number of replicates or treatments which can reasonably be accommodated.

Addition of small amounts of water saturated with $CO_2$ is also an effective method to gently increase $pCO_2$ (Garzke et al., 2016; Paul et al., 2016; Riebesell et al., 2010; Runge et al., 2016), but when applied to microcosms, must be coupled with semi-continuous techniques to avoid the limitations noted for batch culture techniques described above.


In semi-continuous cultures, a portion of the volume is periodically replaced with fresh media of the appropriate chemistry in order to maintain cultures in exponential growth and semi-static population abundance (Feng et al., 2008; Fu et al., 2007; Hutchins et al., 2007). In addition to being suitable for plankton populations sensitive to mechanical stress, semi-continuous culturing allows for repeated monitoring of chemical and biological factors through use of the volume removed from

experimental vessels upon dilution. Semi-continuous culture employs frequent dilutions, or low biomass concentrations to avoid metabolically driven changes in carbonate chemistry. Resource availability, biomass concentrations and metabolic rates determine the dilution interval, which can vary from daily dilutions to every 4 days (Cripps et al., 2016; Gao et al., 2009; Hildebrandt et al., 2016; Kurihara et al., 2004; Vehmaa et al., 2012; Webb et al., 2016; Xu and Gao, 2015).




Gas exchange has been used as a supplementary control mechanism in microcosms (Müller et al., 2012; Webb et al., 2016),
flow through systems (Fangue et al., 2010) and mesocosms (Garzke et al., 2016; Horn et al., 2016), and as a primary control
mechanism in one mesocosm (Wahl et al., 2015a). The ACCS employs passive gas exchange as a modification to the semi-
continuous method. Culture vessels are maintained in a large volume chamber with a controlled atmosphere between water
changes. This modification can reduce the necessary frequency of water changes and/or increase the allowable biomass
density given an acceptable range of biological perturbation to the carbonate chemistry.

**2 Description of System:**

The main components of the ACCS are (1) gas mixing, (2) gas distribution, (3) gas measurement, (4) equilibration and
atmospheric simulation chambers, (5) verification of $pCO_2$ through discrete measurements (Figure 1).

**2.1 Gas mixing:** The gas mixing system is similar to many others and is described in detail in the supplementary material.
In short, compressed air passes though self-regenerating molecular sieves to strip out $CO_2$, and then flows to a set of mass
flow controllers (MFCs) at constant pressure. Research grade $CO_2$ feeds a second set of MFCs, and the two streams mix to
provide precisely controlled $CO_2$/air gas mixtures to each distribution channel. A set of back pressure regulators maintain a
steady pressure on the downstream side of the MFCs. These are paired with typical two stage regulators on the upstream
side. This ensures a constant pressure drop across the controllers, resulting in steady flow rates.


**2.2 Gas Distribution:** The mixed gas enters a manifold with six adjustable flow meters for each distribution channel. These
allow the user to control and direct the flow of gas to any of the equilibration or atmospheric simulation chambers. Each
distribution channel is equipped with a pressure gauge, which allows the user to ensure that downstream pressure does not
exceed the set point of the backpressure regulators, thereby decreasing the pressure drop across the MFCs.


**2.3 Equilibration and atmospheric simulation:** After humidification, gas mixtures flow to either an equilibration tank or
atmospheric simulation chamber. Pre-conditioning of media is achieved in equilibration tanks by continuous bubbling
through fine pore air-stones. The large atmospheric simulation chambers (120 L volume) are constructed of clear acrylic
with inlet and outlet lines and a large access hatch. They can be maintained under static or varying temperature and light
conditions. The controlled atmosphere supplied to these chambers allows gas exchange to offset metabolically driven
changes in water chemistry in planktonic cultures that will not normally tolerate direct bubbling.

Using semi-continuous techniques, numerous culture vessels can be maintained in each chamber. Alternatively, for
populations that are not sensitive to mechanical stress, direct bubbling is possible. A flow control manifold placed in the





chamber distributes gas from the chamber inlet into the culture vessels. Intermediate configurations including both direct bubbling and passive gas exchange are also possible, maximizing efficient use of the gas streams.

**2.4 Verification of $pCO_2$:** The $pCO_2$ of the gas entering the manifold or exiting each of the distribution channels can be quantified (Licor 820). Gas entering the $CO_2$ sensor is dehumidified by passing through a nafion drying tube supplied with a
counter flow around the tube of the dry air from the compressor.

Carbonate chemistry parameters are measured frequently by subsampling cultures or using the displaced volume from dilutions. Here, we use daily spectrophotometric pH (Dickson et al., 2007) and total dissolved inorganic carbon (DIC) measurements (Apollo Sci-Tech), supplemented by periodic open cell Gran titrations for total alkalinity (Dickson et al., 2007) and calculate the remaining parameters using CO2sys (Lewis and Wallace, 1998). Details of these methods are
available in the supplementary materials.

**3 Gas Exchange Effects**

The primary aspect that differentiates the ACCS from other OA experimental approaches is the prominent use of controlled atmospheres. Movement of $CO_2$ across the air/water interface in culture vessels in the ACCS helps keep pH and $pCO_2$ closer to target values than in a simple batch or semi-continuous approach. While this is not very clear when comparing across
studies with different organisms, volumes and $pCO_2$ (Table 1), it can be demonstrated by the effect of surface area to volume on change in pH in batch cultures.

When square cross section flasks are filled to 90%, 70% and 50% of capacity, surface area to volume ratios increase accordingly (90, 116 and 162 $cm^2 L^{-1}$ respectively). As photosynthesis draws down $CO_2$ in batch cultures of *Emiliania huxleyi*, flasks with the highest surface area to volume exhibit the smallest departures in pH from the media blanks (Figure
2). This effect is most pronounced when there is a large gradient across the surface, as in the later stages of these batch cultures. Cell density in these cultures exceeds $1x10^5$ cells $ml^{-1}$ on day 3 and $8x10^5$ cells $ml^{-1}$ on day 5. Additional testing in the ACCS showed that under cell densities typical in our semi-continuous culture (about $1x10^5$ *Emiliania huxleyi* cells ml-1), carbon drawdown was reduced by 20 to 30 percent by passive gas exchange in flasks open to the atmospheric simulation chamber, compared to cultures that were kept capped.

**4 Performance**

This system is capable of maintaining a large number of plankton cultures close to their target $pCO_2$ conditions. Carbonate chemistry is steady over time, and treatment differences are stable, which allows cultures to acclimate to given conditions for many days (Figure 3). Semi-continuous cultures do show some effect of biological modification of the mean $pCO_2$ values, which are reduced in algal cultures and increased in zooplankton cultures relative to the pre-equilibrated media. Despite





these effects, the ACCS maintains strong treatment separation (Figure 4). The average difference between the pre-equilibrated media and cultures after 24 hours of growth was 75 µatm $CO_2$ during the experiment illustrated in Figure 3. These measurements represent the greatest departure of conditions from the pre-equilibrated media because samples were taken just before dilution with fresh media. Average culture conditions over 24 h are likely closer to that of the pre-equilibrated media. Details for the 4 experiments outlined in this study are reviewed in Table 2. These experiments are a

representative subset of more than a dozen experiments which have been carried out in the ACCS.

Use of the ACCS with semi-continuous culture and passive gas exchange allows a diurnal cycle to develop in cultures. During the light period, $pCO_2$ drops in algal cultures, and rises with net respiration during dark periods. Experiments in which we monitored carbonate chemistry in detail over 24 hours showed that during an 8-hour dark period, $pCO_2$ increased

on average 10-20 uatm in cultures of Emiliania huxleyi, and average drawdown over an 8-hour light period were 30-60 uatm $CO_2$. This corresponds to a light period change in pH of about 0.03 and about 0.4 % change in DIC. These differences are consistent in magnitude with the increases in $pCO_2$ for passively maintained mixed cultures of copepods and their prey over 24 hours in the ACCS (Table 1). These changes in pH of between 0.02 and 0.05 pH units, correspond to $pCO_2$ increases between 50 and 100 ppm $CO_2$ over the course of 24 hours. The daily dilution ensures that this cycle in $pCO_2$ remains

centered around a steady average value and is not overwhelmed by increased biomass or metabolic activity in the culture vessels over time.

This level of diurnal variability is comparable to that found in some of the more stable natural systems. Hofmann et al. (Hofmann et al., 2011) deployed pH sensors in a range of habitats and found short term variability of about 0.02 in open

ocean environments, and ranging up to greater than 0.5 in some estuarine and reef habitats. This variability can be driven by mixing, advection, upwelling, and community metabolism. Biomass density and water residence time are two major factors which govern the magnitude of pH variability in natural settings and which can be approximated in semi-continuous culture.

Metabolically driven changes in microcosm carbonate chemistry can be large when allowed to go unchecked. For instance,

in the case of Villafane (2015) rapid growth rates in their high $pCO_2$ treatment resulted in increases in pH of about 0.4 units per day after they were reset to a nominal value each morning. This level of variability rivals that of some of the most dynamic marine systems. Dilution of cultures is a rough analogue of water exchange or residence time in a given habitat. When water changes or dilutions have a longer time interval, a change in pH of 0.15 to 0.5 between water changes is common (Table 1). This corresponds to approximately a 400 – 800 µatm difference in $pCO_2$, which rivals or exceeds the

difference between treatments in many experiments.

In the ACCS, active control of biomass and turnover of water through dilution is coupled with gas exchange to keep the variability within realistic bounds, and to enhance the realism of the experimental system. When biomass in culture and the

surface area to volume ratios are moderate, the stability of pH in the ACCS is comparable to other experiments using semi-
continuous culturing but no subsequent gas exchange (Table 1). When operated with care and attention to the considerations
detailed below, very tight carbonate chemistry control is possible (e.g. experiment 4). All semi-continuous methods can
modify the variability in carbonate conditions by changing the biomass concentration in culture. The ACCS can also
increase or decrease variability by modulating the effects of gas exchange on the cultures (Figure 2).

## 5 Considerations

Water vapor can exchange across the surface as well as $CO_2$, resulting in evaporation if air in the atmospheric simulation
chamber is dry. In order to humidify air, gas streams can pass through membrane based nafion humidifiers or more simply,
air entering the chamber can be bubbled though a column of deionized water. Water changes associated with semi-
continuous culturing also help maintain appropriate salinity levels.

The development of a substantial diffusive boundary layer at the air water interface will reduce the rate of gas exchange.
Mixing of both the air and water minimizes this effect. Air circulation in the atmospheric simulation chambers is driven by
inflow of gas at about 5 L min-1. Automatic stirring of culture vessels that is sufficiently gentle is difficult to achieve in an
open vessel and manual stirring requires that the chamber be opened. Early versions of the ACCS incorporated integral
gloves for handling vessels without opening the chamber, but they were cumbersome. Therefore, culture vessels are not
actively stirred, and boundary layers may develop between mixing of cultures upon feeding or dilution.

Air in the chamber is replaced with the room air when opened, and takes some time to be replenished with treatment air.
This is particularly important when the chambers are located in an enclosed space such as a walk-in incubator. Lab air in
general, and in enclosed spaces where people are working in particular, can quickly achieve very high $pCO_2$. Good general
ventilation, monitoring of room air $pCO_2$ and fans that can be activated before chambers are opened is recommended.
Scrubbing of room air is also possible, but we found ventilation to be simpler and more effective. The performance of this
system has improved over time with the improvements in ventilation and sample handling. When operated with care, it
compares favorably to other published results from semi-continuous culturing studies (Table 1).

## 6 Advantages

Advantages of this system include the ability to allow diurnal cycles in carbonate chemistry to develop, but also the ability to
modulate those effects through gas exchange, as in near shore environments. Depending on the organismal metabolic rates,
semi-continuous dilution volumes, and the surface area to volume ratio of the culture vessels, the diurnal cycle can be tuned
such that it is similar in magnitude to that in natural systems. The dilution of cultures and gas exchange keep carbonate
chemistry close to target values while still allowing relatively dense cultures. It has cost effective gas handling, flexibility,





and is suitable for delicate planktonic organisms, including holo- and meroplankton. Vessel $pCO_2$ is gently controlled via passive gas exchange, or more robust cultures may be bubbled directly.

A key feature of the ACCS is the ability to accommodate a large number of culture vessels. For example, tracking copepod egg production of individual females, or the hatching success and development of egg clutches requires more culture vessels than can be accommodated by most OA experimental methods. The ACCS enables analysis of these metrics, or the tracking

of individual breeding crosses or families over time. These kinds of studies get at the variability of responses within a population which is important both in data interpretation and in predicting how adaptation mediates OA response.

The addition of backpressure regulators can increase the reliability of mixed gas concentrations. MFCs need a constant pressure drop across the controller to operate correctly. Many system operators do not control pressure downstream of MFCs, and may adjust upstream pressure to counteract pressure changes in the distribution system (Falkenberg et al., 2016).

It is imperative that the concentrations of the gas supply remain steady regardless of the state of the airstones in the pre-equilibration tanks or small flow adjustment which can cause cascading effects through pressure changes. The backpressure regulators in the ACCS maintain a pressure of 15 psi on the MFC outlets as long as the downstream pressure does not exceed that value. When paired with 2 stage regulators upstream, which maintain a pressure of 30 psi on the MFC inlets, the back pressure regulators improve MFC performance. The addition of backpressure regulators would benefit any system which

uses MFCs.

## 7 Experimental design considerations

Many OA studies have not fully addressed the independence of their replicates in the design or the description of experimental systems (Cornwall and Hurd, 2015). While this system is not immune to these issues, some steps can be taken in its use to minimize these concerns. With six distribution channels for each concentration, one could supply three

independent equilibration tanks paired with 3 atmospheric simulation chambers for each $pCO_2$ treatment (similar to tank array b in Cornwall and Hurd). A nested design would then be appropriate for statistical analysis of results from culture vessels in those chambers. Alternatively, the tanks used for equilibration could be randomly reassigned to different treatments when they are refilled each day, reducing potential issues arising from using a common equilibration tank for all replicates in each treatment. Similarly, atmospheric simulation chambers can be easily reassigned to different treatments

through the course of the experiment. Preliminary tests in the system showed no effect of atmospheric chamber on algal cultures, but contamination or other problems could differentially affect the chambers and confound results if chamber and treatment conditions are identical. The greater the number of experimental conditions, the less flexibility there is to increase independence of experimental units (culture vessels in this case). For instance, if the design requires three temperatures for each of 3 $pCO_2$ levels, then equilibration can be carried out inside the atmospheric simulation chamber, and 2 chambers can

be assigned to each $pCO_2$/Temperature condition.

## 8 Conclusion

The flexible design of the ACCS has proven to be effective in experiments with a wide variety of organisms, including bacterioplankton, phytoplankton , micro- and meso- zooplankton, and larval invertebrates (Buckham, 2015; Christmas, 2013; Kendall, 2015; McLaskey et al., 2016; Siu et al., 2014; Still, 2016; Wuori, 2012). It is particularly designed for the maintenance of organisms which are sensitive to bubbling, and for increasing the realism which can be achieved in a microcosm through diurnal fluctuations and natural gas exchange processes.

### Data Availability

All data are either available in the supplementary material (data for Figure 2) or at the BCO-DMO data repository.

### Author Contribution

B. Love was the lead on design and construction of the system, supervision and execution of the initial testing, system improvements maintenance and trouble shooting. M.B. Olson collaborated on these same activities. T. Wuori carried out initial testing, including surface area to volume experiments. B. Love drafted the manuscript with primary editorial assistance from M.B. Olson and helpful commentary from T. Wuori.

### Competing Interests

The authors declare that they have no conflict of interest.

### Acknowledgements

We are grateful for the support of the faculty and staff of the Shannon Point Marine Center, for accommodating and enhancing the ever expanding scope of the ocean acidification lab. Scientific Technical Services built and repaired countless devices for this project. Several undergraduate students contributed through research projects, including Sabelo Duley, Daniel Hernandez, Julius Allison, and Daniel Hodge. These students were supported by the NSF Research Experience for Undergraduates and Multicultural Initiative in Marine Science Undergraduate Programs. This work was funded through National Science Foundation awards OCE 0961229 and OCE 1220664.



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




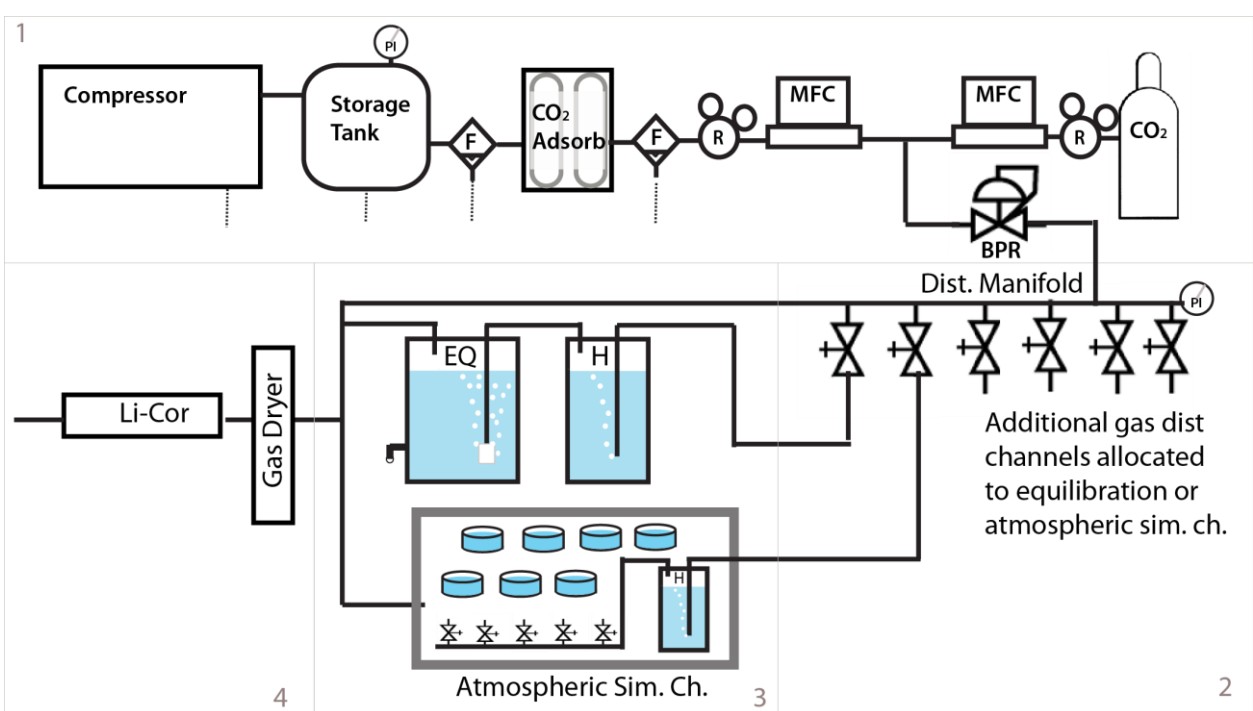

**Figure 1:** A general schematic of the system including gas mixing (1), gas distribution (2), equilibration tanks and atmospheric
simulation chambers with temperature control (3), and verification of $pCO_2$ (4). A pair of MFCs is present for each $pCO_2$
treatment. The components in 2 and 4 are also repeated for each treatment for up to 9 total equilibration tanks and up to 9
atmospheric simulation chambers. Abbreviations in the schematic include pressure indication (PI), self-draining filter (F),
regulator (R), back pressure regulator (BPR), humidifier (H), and equilibration tank (EQ). The distribution manifold is plumbed
with flow meter valves so each gas distribution channel can be visually checked and adjusted for flow. Dotted lines indicate drain
lines.



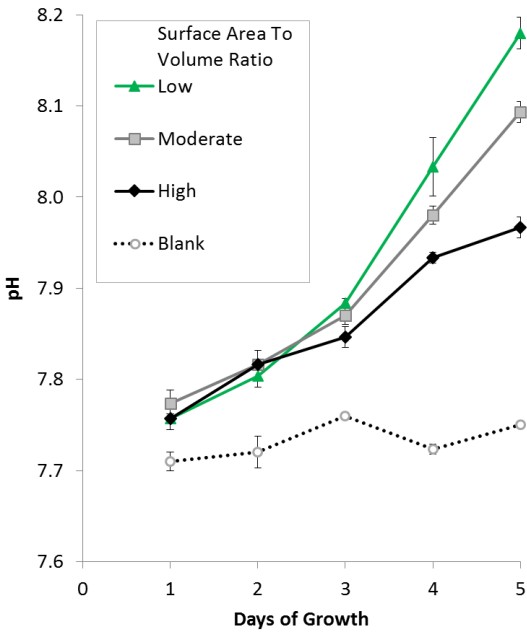

**Figure 2:** **Flasks with high surface area to volume ratios show more evidence of atmospheric equilibration compared to low surface area to volume flasks when biomass and air/water $CO_2$ gradients are high after several days of growth. Treatments represent triplicate batch cultures of Emiliania Huxleyi. Surface area to volume ratios tested are 90 (low), 116 (moderate) and 162 (high) cm$^2$ L$^{-1}$, corresponding to volumes of 900, 700 and 500 ml in a square 1 liter flask (n=3). Blanks, which do not contain algal cells retain relatively constant pH.**






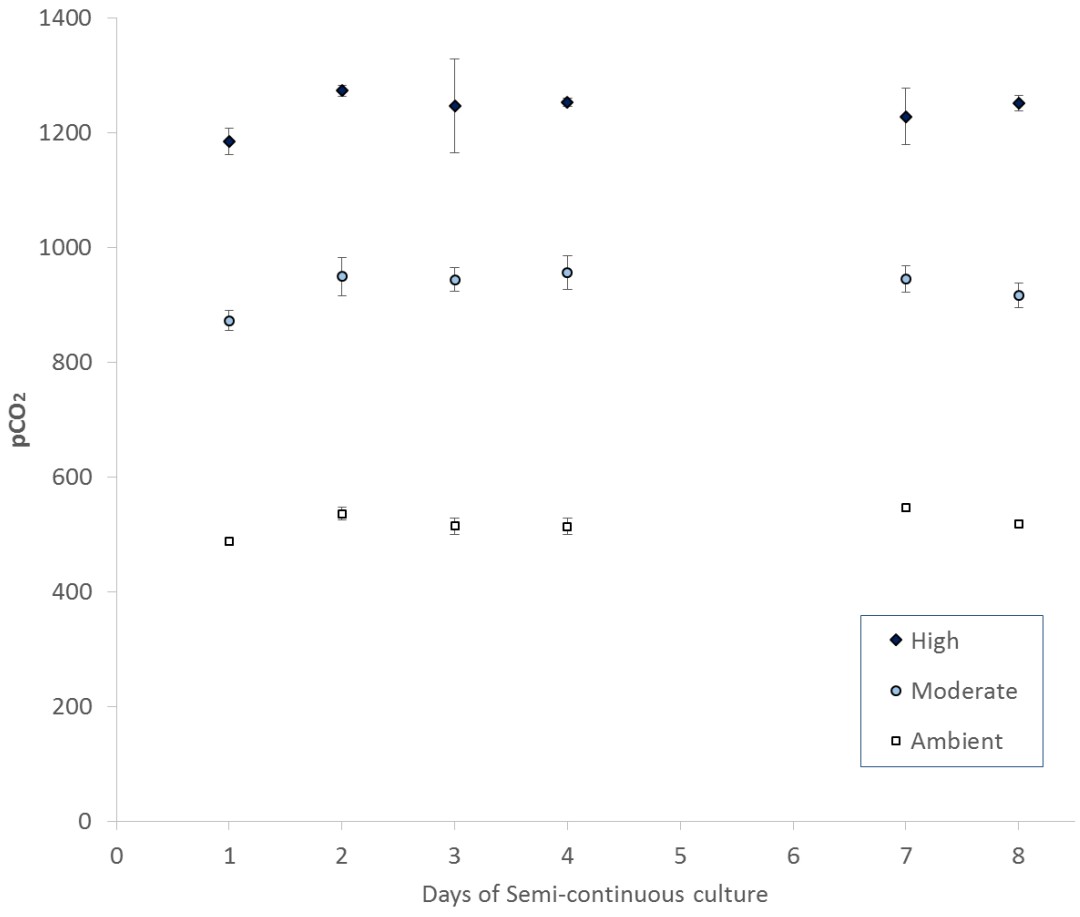

**Figure 3 – Maintenance of pCO₂ over time in mixed cultures of the copepod *Calanus pacificus* and its prey *Ditylum brightwellii***
**(n=3, Experiment 2). Cultures were maintained in atmospheric chambers under semi-continuous culture with artificial atmospheres with no bubbling and media for daily dilutions was bubbled with air at 400 (ambient), 800 (moderate) and 1200 (high) μatm pCO₂**




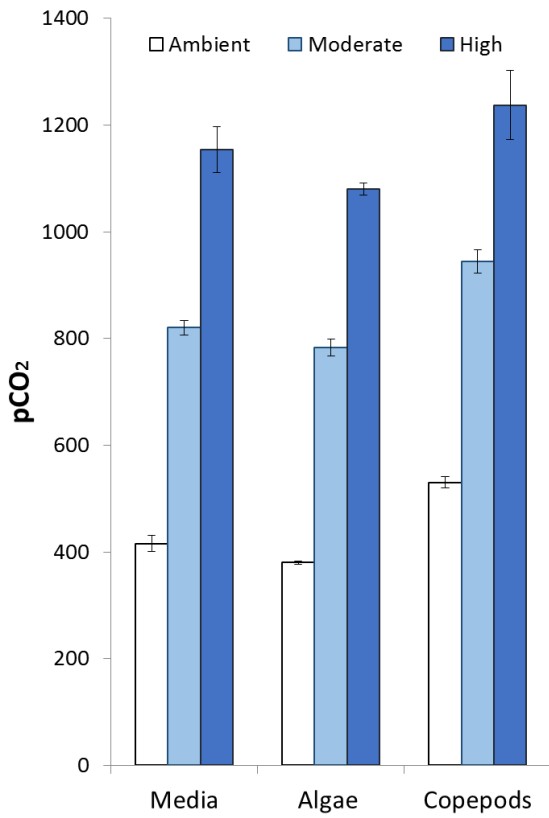

**Figure 4:** **System performance as measured by maintenance of pCO₂ in pre-conditioned media, algal cultures (diatom *Dytilum brightwellii*), and mixed cultures of the algae and the copepod *Calanus pacificus* (n=6, Experiment 2). Algae and Copepods were under semi-continuous culture with artificial atmospheres and media for daily dilutions was bubbled with air at 400 (ambient), 800 (moderate) and 1200 (high) µatm pCO₂.**






**Table 1: Comparison of daily variability in pH in semi-continuous OA studies without direct bubbling. $\Delta$ *pH* is the mean absolute difference between pH before and after dilution (where available). Dilution interval indicates days between dilution of cultures. Ambient conditions are nominally between 380 and 450 $\mu$atm $CO_2$. High conditions are nominally between 1000 and 2000 $\mu$atm $CO_2$. All studies apart from ACCS experiments are simple semi-continuous cultures with closed conditions between dilutions.**

| Study | Organism | Dilution Interval (Days) | $\Delta$ pH Ambient | $\Delta$ pH High |
|---|---|---|---|---|
| Webb 2016 | Coccolithophore | 4 | 0.45 | 0.55 |
| Hildebrandt 2016 | Copepod | 4 | 0.19 | 0.13 |
| Xu and Gao 2015 [a] | Coccolithophore | 3 | 0.07 | 0.07 |
| Kurihara 2004 | Copepod | 2 | 0.03 | 0.15 |
| Gao 2009 [a] | Coccolithophore | 1 | 0.08 | 0.08 |
| Vehmaa 2012 | Copepod | 1 | 0.20 | 0.06 |
| Cripps 2014 | Copepod | 1 | 0.03 | 0.01 |
| **ACCS Exp 1 [b]** | **Copepod** | **1** | **0.09** | **0.04** |
| **ACCS Exp 2** | **Copepod** | **1** | **0.05** | **0.02** |
| **ACCS Exp 3** | **Copepod** | **1** | **0.07** | **0.02** |
| **ACCS Exp 4** | **Copepod** | **1** | **0.02** | **0.001** |

[a] maximum drift as reported, not specified by treatment

[b] 75% of difference between of pH of media used to dilute the cultures and pH at end of dilution interval as an estimate of $\Delta$pH when a 75% replacement of volume is used for experiments 1-4.



**Table 2: Details of recent experiments using the ACCS. Each experiment includes an algal prey species (listed first) and a copepod**
**(listed second). All cultures were maintained either with direct bubbling (active) or semi-continuous culture in atmospheric simulation chambers (passive). Algal prey cells were added to the copepod cultures twice daily (every 12 hours) to maintain saturating feeding conditions.**

| Experiment | Species | $pCO_2$ Control | T (°C) |
|---|---|---|---|
| 1 | *Prorocentrum micans* | Passive | 12 |
|   | *Calanus pacificus* | Passive |    |
| 2 | *Ditylum brightwellii* | Active | 12 |
|   | *Calanus pacificus* | Passive |    |
| 3 | *Rhodomonas salina* | Active | 12 |
|   | *Acartia hudsonica* | Passive |    |
| 4 | *Rhodomonas salina* | Active | 17 |
|   | *Acartia hudsonica* | Passive |    |
