# Peer review of "Technical Note: A minimally-invasive experimental system for $pCO_2$ manipulation in plankton cultures using passive gas exchange (Atmospheric Carbon Control Simulator)"

_Biogeosciences, 2016_

## Referee Comment (RC1) · Anonymous Referee #1 · 4 Jan 2017

General Comments

This is a very interesting and well-written manuscript that describes a sophisticated and novel ocean acidification (OA) simulation system. Of particular interest is the use of CO2-enriched headspace to compensate for the CO2 drawdown effects of photosynthetic organisms. In addition, the opportunistic use of fluctuations in carbonate chemistry (caused by water replacement) to mimic diurnal changes in carbonate chemistry in the field is interesting, especially in the context of the ever-increasing demand for more realistic OA simulation systems.

[Figure]

The authors acknowledge that the design of this system is vulnerable to the pseudoreplication outlined in Cornwall and Hurd (2015). However, insufficient detail is given on how this can be compensated for during data analysis, e.g. does a 'nested design' fully counter these problems? Have the authors considered evaluating the variability caused by 'Culture Vessel' as a random effect? This can quantify the independence or non-independence of phytoplankton cultures maintained in the same chamber.

While this manuscript is very well-written, the overview of the system (Section 2) lacks a straightforward and thorough description of the simulation apparatus. I am still unclear on how large the culture vessels are, how they are physically positioned in the simulation chamber, and how the chambers are maintained from day to day. The introductory passage in Section 2 should be expanded to include a detailed description.

Finally, as this manuscript describes a novel OA simulation system, I expected the authors to include all measurements used in the validation process, i.e. Total Carbon and Total Alkalinity data. The supplementary document provides ample detail on how these measurements were taken, but does not include the data needed to gauge the success of the system. In any case, these measurements should be included in the manuscript of any description of a novel OA simulation system.

Specific Comments

Line 24-25: 'Manipulative OA research has increased dramatically...' Comment: On what timescale?

Lines 41-42: 'The ACCS is designed to accommodate differing trophic levels and differing levels of desired carbonate control, re-equilibration, or metabolically driven cycling.' Comment: I recommend that the authors briefly expand on these claims in this passage, allowing the reader to place the system in the context of other acidification apparatus.

Line 67: In the unpublished data referred to here, what was the control, i.e. the unbubbled culture, that the authors compared with the bubbled culture? Was this part of a validation exercise? If so, please expand on this statement and refer to the validation trial.

Line 176-177: 'This level of variability rivals that of some of the most dynamic marine systems.' Comment: This statement needs an example and citations.

Technical Comments

Line 55: 'e.g.' is in the middle of the in-text citation.

Line 73: 'e.g.' is in the middle of the in-text citation.

Figure 1: Add an explanation of the abbreviation 'sim.ch.'

Table 1: Please specify that pH is reported on the Total Hydrogen Ion Scale.

Figures 3 and 4 are mentioned in the text before Figure 2.

Figure 3: Are the error bars SD or SE?

Figure 4: Are the error bars SD or SE?

Figure 2: Are the error bars SD or SE?

Line 160: Scientific name (Emiliania huxleyi) is not italicised.

Lines 168-169: There is a formatting error in the citation for Hofmann et al. (2011).

---

## Author Comment (AC1) · 4 Jan 2017

We thank the reviewer for the thoughtful suggestions. We agree that the main comments that the reviewer makes would all improve the manuscript and will be happy to make those changes. In particular:

1) Add a more detailed description of the system, in particular the environmental chambers. We were attempting to keep everything as concise as possible, in the interest of complying with the short communication/technical note format. We can certainly expand the description somewhat and it is good to have the reviewers comments to guide

which areas most need a more detailed description. If other readers or reviewers have additional points in the system description that could be presented more clearly, please let us know.

2) Statistical treatment/pseudoreplication discussion can be more complete. The suggestions here are also very cogent and achievable. More details on statistical approach to minimize the limitations of the experimental design can be added. While we have been aware of these issues from the start, tools and consensus on how to address these concerns have been emerging over the time we have been building and refining the system. The suggestion of including the chamber as a random effect is more clear way of expressing the treatment that we were suggesting with the nested design. We will incorporate that into our protocols for data analysis in this section.

3) The reviewer would like us to include a more complete suite of carbonate chemistry parameters. We chose not to do this initially because were attempting to compare the system to the performance of other approaches and pH was the parameter that was universally available to make that comparison. Again, in the interest of keeping it short, we did not include additional parameters. We do, of course, have those data, and would be glad to add them to the manuscript.

We find that there are several different ways to present these kind of data, which tell different stories about the system. In our existing data presentation, we highlight the diurnal differences in pH in the system. Perhaps a data table showing means with variability over the course of several entire experiments for the whole suite of carbonate parameters would be the best approach for further data summary. Time series plots of these parameters over time can also be useful, but again, we don't wish to unduly expand the manuscript beyond the modest footprint of this format. If readers and reviewers have a preference for time series plots versus summary data tables, we would welcome that feedback as well.

---

## Referee Comment (RC2) · Anonymous Referee #2 · 8 Mar 2017

General comments: The manuscript is well structured and clearly written. This novel system is well thought out and I believe it to be an important contribution to the task of ocean acidification studies. The introductory material presents well the advantages and limitations of various approaches. Also the system is described in a sufficient way. Although, this is the microcosm study, which has various limitations, the authors acknowledge and describe most of it is limitations. Despite that, I think there is also one limitation of this study, which should be mentioned. These are the factors which affect the gas transfer velocity, hence gas exchange with the atmosphere. As this is closed system, in reality the gas exchange with atmosphere could be (and most likely is) quite

distinct from what it could be observed in microcosms. As in reality it is affected by wind speed, convection etc. , lack of these factors in microcosms may limit the realism of the experiment. Beside that, also, are ions monitored as well? The ionic strength also affects CO2. And I think it should also be added, what is the volume of each vessel?

I particularly appreciate the possibility of using different, controlled atmospheres. This could be especially useful in stimulating past and/or future ocean acidification. This would be helpful in studies of the carbonate chemistry of seawater under past, present and future conditions, especially under ongoing climate change.

Specific comments:

in pCO2 be p should be italicised

Line 18: Although for me it is clear what OA stands for, I think it should be explained anyway

Line 129: the lower case of 2 in CO2 sys. And please explain: CO2sys is the abbreviation for?

Line 143: -1 should be upper case

Line 155: "more than a dozen": I think it is better to put the actual number of experiments than just the word "dozen"

Line 160, 419: Emiliania Huxleyi should be italicised

Line 167: should be just:" Hoffmann et al (2011).

Fig. 1 All the abbreviations should be explained, for example Sim. Ch. or Dist.

Fig. 3, fig. 4: pCO2 is missing units

Supplementary note: Line 23: "minimal pressure": what is the value of this minimal pressure?

---

## Author Response (AR1)

[revised manuscript text omitted]

We are pleased to provide this record of our response to the reviewer suggestions. They were helpful and relatively simple to
address. Thank you for your time in evaluating these changes and your help in getting this manuscript into print. Line numbers
here refer to those in the final manuscript, not the marked up version.

Anonymous Referee #2

 General comments: The manuscript is well structured and clearly written. This novel
system is well thought out and I believe it to be an important contribution to the task of ocean acidification studies. The
introductory material presents well the advantages and limitations of various approaches. Also the system is described in a
sufficient way. Although, this is the microcosm study, which has various limitations, the authors acknowledge and describe
most of it is limitations. Despite that, I think there is also one limitation of this study, which should be mentioned. These are
the factors which affect the gas transfer velocity, hence gas exchange with the atmosphere. As this is closed system, in reality
the gas exchange with atmosphere could be (and most likely is) quite distinct from what it could be observed in microcosms.
As in reality it is affected by wind speed, convection etc. , lack of these factors in microcosms may limit the realism of the
experiment.

> *We appreciate the generally positive response of reviewer two. You make a good point about gas transfer velocity in*
> *a closed vessel such as this one. We have added some text addressing this point in the "gas exchange effects" section*
> *on lines 158-162*

Beside that, also, are ions monitored as well? The ionic strength also affects CO2. And I think it should also be added, what is
the volume of each vessel?

> *We did monitor salinity and that is included in the table summarizing the carbonate chemistry in more detail as*
> *requested by reviewer 1. We give the volume of the atmospheric chambers on line 113, so I think you are asking*
> *about the vessels used in culturing for the example studies here. We have added on line 120-125 a statement that*
> *culture vessel size can vary quite a bit in the system and put in specifics about what vessels we used in the caption to*
> *table 2 where experimental details are laid out.*

I particularly appreciate the possibility of using different, controlled atmospheres. This could be especially useful in stimulating
past and/or future ocean acidification. This would be helpful in studies of the carbonate chemistry of seawater under past,
present and future conditions, especially under ongoing climate change.

Specific comments: in pCO2 be p should be italicised

> *We have made this correction throughout.*

Line 18: Although for me it is clear what OA stands for, I think it should be explained anyway

*We replaced the abbreviation with the full term for the abstract here.*

Line 129: the lower case of 2 in CO2 sys. And please explain: CO2sys is the abbreviation for?

*We have made this correction (Line 142)*

Line 143: -1 should be upper case

*We have made this correction - superscript*

Line 155: "more than a dozen": I think it is better to put the actual number of experiments than just the word "dozen"

*There have been a number of student experiments and repeated experiments or subsets of long runs etc… so it is hard*
*to put a hard number on this, but I made a reasonable estimate. (Line 172)*

Line 160, 419: Emiliania Huxleyi should be italicised

*We have made this correction*

Line 167: should be just:" Hoffmann et al (2011).

*We have made this correction (Line 184)*

Fig. 1 All the abbreviations should be explained, for example Sim. Ch. or Dist.

*We have made this correction by replacing abbreviated text with the non-abbreviated version in the figure.*

Fig. 3, fig. 4: pCO2 is missing units

*We have corrected this.*

Supplementary note: Line 23: "minimal pressure": what is the value of this minimal pressure?

*This has been altered to read "slowly (about 10 ml/min) because with syringe filtering, although the aim to keep*
*pressure low, we do not measure the pressure applied.*

Anonymous Referee #1

General Comments: This is a very interesting and well-written manuscript that describes a sophisticated and novel ocean acidification (OA) simulation system. Of particular interest is the use of CO2-enriched headspace to compensate for the CO2 drawdown effects of photosynthetic organisms. In addition, the opportunistic use of fluctuations in carbonate chemistry (caused by water replacement) to mimic diurnal changes in carbonate chemistry in the field is interesting, especially in the context of the ever-increasing demand for more realistic OA simulation systems.

The authors acknowledge that the design of this system is vulnerable to the pseudoreplication outlined in Cornwall and Hurd (2015). However, insufficient detail is given on how this can be compensated for during data analysis, e.g. does a 'nested design' fully counter these problems? Have the authors considered evaluating the variability caused by 'Culture Vessel' as a random effect? This can quantify the independence or non-independence of phytoplankton cultures maintained in the same chamber.

We appreciate this clarification.  The inclusion of the atmospheric chamber as a random effect is what we meant by a nested design, wherein the atmospheric simulation chamber is "nested" as a variable within each treatment condition.   We have made this more clear in the manuscript on lines 252-254.

While this manuscript is very well-written, the overview of the system (Section 2) lacks a straightforward and thorough description of the simulation apparatus. I am still unclear on how large the culture vessels are, how they are physically
positioned in the simulation chamber, and how the chambers are maintained from day to day. The introductory passage in Section 2 should be expanded to include a detailed description.

*We have included more specific details about the dimensions of the chambers, as well as the volume.  (Line 113) and more details about the size and number of culture vessels that were used in these studies.  We also added a brief description of daily operation including checking the pCO2 level in gas streams, physical handling during dilutions*
*and replacement in the chambers.*

Finally, as this manuscript describes a novel OA simulation system, I expected the authors to include all measurements used in the validation process, i.e. Total Carbon and Total Alkalinity data. The supplementary document provides ample detail on how these measurements were taken, but does not include the data needed to gauge the success of the system. In any case, these measurements should be included in the manuscript of any description of a novel OA simulation system.

*Yes, in the interest of brevity we did not include those data, but we are happy to do so and have included a table with the relevant detailed data in table S1.*

Specific Comments Line 24-25: 'Manipulative OA research has increased dramatically. . .' Comment: On what timescale?

Good catch.  We have added "over the last decade here".

Lines 41-42: 'The ACCS is designed to accommodate differing trophic levels and differing levels of desired carbonate control,
re-equilibration, or metabolically driven cycling.' Comment: I recommend that the authors briefly expand on these claims in this passage, allowing the reader to place the system in the context of other acidification apparatus.

*We have added a bit more detail to each of these claims  - referring to the flexibility in the culture vessels and the ability to manipulate the surface area to volume ratios to influence the gas exchange.*

Line 67: In the unpublished data referred to here, what was the control, i.e. the un bubbled culture, that the authors compared
with the bubbled culture? Was this part of a validation exercise? If so, please expand on this statement and refer to the validation trial.

*We have changed the wording here to: "We observed possible detrimental effects of bubbling, including reduced growth rates and formation of aggregates during culturing of autotrophic dinoflagellates which were resolved when cells were cultured under passive gas exchange conditions (unpubl. data)."  This should indicate that this was an*
*informal observation, not a formal trial.  The surface area to volume expleriment described elsewhere in the manuscript were part of our validation trials.*

Line 176-177: 'This level of variability rivals that of some of the most dynamic marine systems.' Comment: This statement needs an example and citations.

*We have added a reference to Hoffman et al (2011) here and list the observed pH ranges in that study for upwelling*
*and estuarine systems.*

Technical Comments Line 55: 'e.g.' is in the middle of the in-text citation. Line 73: 'e.g.' is in the middle of the in-text citation.

*These have been corrected*

Figure 1: Add an explanation of the abbreviation 'sim.ch.'

*Done*

Table 1: Please specify that pH is reported on the Total Hydrogen Ion Scale.

*Done*

Figures 3 and 4 are mentioned in the text before Figure 2.

*It appears that the reviewer missed the first reference to figure 2 in the gas exchange section. We think that the*
*figures are presented in the correct order.*

Figure 3: Are the error bars SD or SE?

*They are SD. This has been added to the caption.*

Figure 4: Are the error bars SD or SE?

*They are SD. This has been added to the caption.*

Figure 2: Are the error bars SD or SE?

*They are SD. This has been added to the caption.*

Line 160: Scientific name (Emiliania huxleyi) is not italicised.

*Fixed*

Lines 168-169: There is a formatting error in the citation for Hofmann et al. (2011).

Fixed